# Piperacillin/Tazobactam and Meropenem Use Increases the Risks for Acute Graft Rejection Following First Kidney Transplantation

**DOI:** 10.3390/jcm11102726

**Published:** 2022-05-11

**Authors:** Dayana Nasr, Mahmoudreza Moein, Stephanie Niforatos, Sandy Nasr, Mulham Ombada, Farzam Khokhar, Myera Shahnawaz, Bhavya Poudyal, Maroun Bou Zerdan, Dibyendu Dutta, Reza F. Saidi, Seah H. Lim

**Affiliations:** 1Department of Medicine, SUNY Upstate Medical University, Syracuse, NY 13210, USA; nasrd@upstate.edu (D.N.); niforats@upstate.edu (S.N.); nasrs@upstate.edu (S.N.); ombadam@upstate.edu (M.O.); khokharf@upstate.edu (F.K.); shahnawm@upstate.edu (M.S.); poudyalb@upstate.edu (B.P.); bouzerdm@upstate.edu (M.B.Z.); dibyendudutta12@gmail.com (D.D.); 2Department of Surgery, SUNY Upstate Medical University, Syracuse, NY 13210, USA; moeinm@upstate.edu (M.M.); saidir@upstate.edu (R.F.S.); 3Division of Hematology and Oncology, SUNY Upstate Medical University, Syracuse, NY 13210, USA

**Keywords:** broad-spectrum antibiotics, kidney transplant, acute graft rejection, obligate anaerobic organisms, risk factors

## Abstract

Many broad-spectrum antibiotics (BSA) alter the intestinal microbiome that regulates adaptive immune responses. We hypothesized that BSA use before and early after kidney transplant may affect acute graft rejection (AGR). We carried out a retrospective cohort study on all patients who underwent kidney transplants in our institution. Patient demographics, clinical data, diagnosis, and treatment history were collected. Antibiotic use within 2 months prior to transplant and during the hospital admissions for transplant, as well as antibiotic types were recorded. A total of 357 consecutive first transplants were included for analysis. Median age was 52 years (range 7–76). A total of 67 patients received living donor and 290 deceased donor kidneys. A total of 19 patients received BSA within two months prior to transplant and 55 patients during the hospital admission for the transplant. With a median follow-up of 1270 days, 38 episodes of biopsy-proven AGR were recorded. There was no difference in the AGR rates during the first year between patients who received BSA and those who did not. However, the use of piperacillin/tazobactam or meropenem (PM) was associated with increased risks for the development of AGR, irrespective of the source of the donor grafts. Time to development of AGR was also shorter. Our data, therefore, suggest that the use of PM BSA prior to and immediately after kidney transplant increases the risks for AGR.

## 1. Introduction

Antibiotics alter the composition of intestinal microbial community [1]. A balanced intestinal microbiome is needed for immune homeostasis, most notably adaptive immunity [2]. Disruption of the balanced microbiome results in immune dysregulation that may either be detrimental or advantageous to the hosts. For example, the use of broad-spectrum antibiotics (BSA) adversely affected the treatment outcome in patients who received immune checkpoint inhibitors for advanced cancer [3,4] and increased risks for the development of acute graft-versus-host disease (GVHD) following allogeneic hematopoietic stem cell transplant (HSCT) [5,6,7]. Some BSA may, however, possess anti-inflammatory properties [8,9,10]. The immunomodulatory effects and anti-inflammatory properties of antibiotics are being exploited for treating certain immune-mediated disorders such as bronchiolitis obliterans following allogeneic HSCT [11] and lung transplant [12], as well as rheumatoid arthritis [13]. Based on these considerations, it is possible that BSA use may affect the outcome of solid organ transplant (SOT). To our knowledge, this potential effect of BSA has never been previously investigated. We hypothesized that BSA use shortly before and soon after transplant affect the risks for development of acute graft rejection (AGR) following kidney transplant. To test this hypothesis, we carried out a retrospective single-center cohort study of patients who underwent first kidney transplant at State University of New York Upstate Medical Center, Syracuse, New York, NY, USA.

## 2. Patients and Methods

### 2.1. Data Collection

A retrospective cohort study was performed on all patients who underwent kidney transplant from 1 January 2016 to 31 December 2020. Patient demographics, clinical data, diagnosis, and treatment history were collected. Continuous variables collected included age, days from transplant to occurrence of AGR, and follow-up periods. Categorical variables collected included gender, donor source, first or subsequent transplant, donor/recipient HLA discrepancy, use of BSA before and during the peri-transplant periods, name of the antibiotic used, and the reason for the prescribing of the antibiotics. Antibiotics were considered broad-spectrum if they were effective against both Gram-positive and Gram-negative bacteria [14]. We chose the 2-month time point before transplant to evaluate the effects of antibiotics on AGR since microbiome changes induced by antibiotics last longer than 2 months [15,16] and previous works in cancer patients treated with immune checkpoint inhibitors showed that BSA use within 2–3 months adversely affected the treatment outcome [3,4]. All patients received a single dose of prophylactic cefazolin in the Operating Room on the day of transplant, and prophylaxis against Pneumocystis jiroveci pneumonia (PJP) with Trimethoprim/Sulfamethoxazole after transplant. In patients who were allergic to sulfa drugs, dapsone or atovaquone was used instead. The study was conducted with exemption from the State University of New York Upstate Medical University Institutional Review Board.

### 2.2. Statistical Analysis

Risks for developing AGR during the first year after transplant were plotted as a time-dependent covariate using the Kaplan–Meier method. Differences in the clinical characteristics between the antibiotic-treated and non-antibiotic groups were evaluated using the Chi-square tests. Differences in the various clinical and laboratory parameters were calculated as medians and compared using the Student’s *t* test. A two-sided *p* value of <0.05 was considered statistically significant and ≥0.05 but ≤0.1 was considered a trend towards significance. The MedCalc softwares (Belgium) were used for data analysis.

## 3. Results

### 3.1. Patients

A total of 402 kidney transplants were performed during the study period. However, only 357 consecutive patients (143 females and 214 males) were included in the study. The other 45 transplants were excluded because they received second or subsequent transplant. Median age of the entire group was 52 years (range 14–76). Deceased donor kidneys were used in 290 and living donor kidney in 67 cases. All patients received standard immunosuppressive agents, and in those with donor-specific antibodies the patients also underwent plasmapheresis and immunoglobulin therapy prior to the transplant.

The characteristics of the patients who did not receive any antibiotics (except for the one dose of intra-operative prophylactic cefazolin and post-transplant PJP prophylaxis) (*n* = 286) and those who received antibiotics (in addition to the one dose of intra-operative prophylactic cefazolin and post-transplant PJP prophylaxis) (*n* = 71) are shown in Table 1. BSA use within two months before transplant was reported only in 19 patients (5%) and during the peri-transplant period in 55 patients (15%). No statistical differences were observed in the age, gender distribution, median follow-up period, and donor/recipient HLA discrepancies between the two groups. However, the proportion of patients who were transplanted with living donor kidneys was significantly higher in the group that did not receive any BSA (21% vs. 10%) (*p* = 0.04).

The BSA administered to patients are shown in Figure 1. The most commonly used BSA was piperacillin/tazobactam, followed by the cephalexin and ciprofloxacin. Probably related to the concern about the possible induction of seizure in patients with renal impairment, none of the patients received imipenem/cilastatin. The reasons for the use of BSA before and immediately after the transplants are shown in Table 2. By far the commonest reasons for the use of BSA pre-transplant were urinary tract infections (7/19) (36.8%) and cellulitis (17.6%), and immediately post-transplants were recipient bacteruria (18/55) (32.7%), donor bacteruria (8/55) (14.5%), donor bacteremia (8/55) (14.5%), and recipient soft tissue infection (7/55) (12.7%).

### 3.2. Acute Graft Rejection and Graft Survival

With a median follow-up of 1270 days from the day of transplant, 26 cases of biopsy-proven AGR were recorded during the first year after transplant. The overall AGR rate during the first year after transplant was 7.3%. There was no difference in the probability of developing AGR during first year after transplant between those who received BSA and those who did not (Figure 2).

We next examined the effects of BSA with strong activities against intestinal obligate anaerobic bacteria. Among the antibiotics shown in Figure 1, two antibiotics were included for analysis: Piperacillin/tazobactam and meropenem. Since the number of patients who received meropenem was small (*n* = 4), we combined these two antibiotics (PM) as a single group. Patients who received PM were at significantly higher risks for developing AGR during the first year after transplant compared to those who did not receive any additional antibiotics (*p* = 0.002) (Figure 3a). The time to AGR development was also shorter in patients who received PM (median 16.5 days, range 8–164 vs. median 139 days, range 87–283) (*p* = 0.06). In contrast, there was no difference in the risks for AGR between patients who received non-PM BSA and those who did not receive additional antibiotics (Figure 3b). To rule out the possibility that the difference in AGR observed with PM use was due to differences in the patient characteristics, we next examined the clinical characteristics of the groups. We did not observe any statistical significance in the age, gender, degrees of HLA matching, and donor source of the graft between those who received PM BSA and those who did not (Table 3), arguing against the higher risks for AGR observed in the PM group were the results of a difference in the patient characteristics.

Since there was a higher number of patients who did not receive BSA (PM or non-PM BSA) underwent transplant with kidneys from living donors, we next analyzed the effects of PM BSA on AGR according to the donor source of the kidney grafts. There was also not any significant difference in the rates of AGR during first year after transplant between those who received deceased donor kidneys and those who received living donor kidneys (6.6% vs. 10.4%, *N.S*.). Similarly, the increased risks for AGR development during the first year after transplant associated with the use of PM BSA were observed in the entire group of patients, irrespective of the source of the donor grafts (Figure 4).

## 4. Discussion

More than 20,000 kidney transplants are performed each year in the United States and the number is increasing [17]. Modern immunomodulatory approaches and the progressive increase in the use of living donor kidneys contribute to the improved graft outcome in these patients. Even so, AGR remains a problem in up to 20% of the transplants [18,19]. Although the immediate graft outcome in most cases is favorable, AGR is associated with increased risks for longer-term graft failure and death from cardiovascular disease and cancer [20]. Investigation into actionable factors that may increase the risks for AGR is, therefore, much needed. Based on the findings that BSA modify the intestinal microbiome and pro-inflammatory cytokine responses [21,22,23], we sought in this study to determine how BSA use might affect the risks for AGR during the first year after kidney transplant.

We chose to examine the effects of the use of BSA within two months before transplant and during the peri-transplant period because we reasoned that the risks for AGR during the first year are likely to be influenced by the immune status of the patients at the early stage following the kidney transplant when a balanced host-versus-graft effects is yet to be established. Furthermore, data on the use of antibiotics during these periods are more accurately captured than after the patients have been discharged from the hospital and sent back to their referring physicians for follow-up. With this approach, we found that BSA use within two months of transplant or during the peri-transplant period was not associated with any significant increase in the risks for the development of AGR during the first year after the first kidney transplant. However, patients who received BSA that included PM were at significantly higher risks for developing AGR, irrespective of the source of the donor grafts, compared to patients who did not receive any additional antibiotics or only non-PM BSA. Alarmingly, piperacillin/tazobactam was by far the most commonly used BSA in this cohort of patients.

Most of the bacterial species that colonize the intestinal tract belong to the phyla *Firmicutes* and *Bacteroidetes*, while lesser phyla include *Actinobacteria*, *Proteobacteria*, and *Verrucomicrobia* [24]. *Faecalibacterium prausnitzii*, a species of the *Firmicutes* phyla, is the most common bacteria in the healthy intestine, forming more than 5% of the bacteria in the intestine. Various studies have suggested the role of *Faecalibacterium prausnitzii*, *Bacteroides thetaiomicron*, *Bacteroides fragilis*, and *Akkermansia muciniphila* in maintaining the intestinal immune homeostasis [25]. For example, the Clostridial species that belongs to *Fermicutes* upregulates intestinal regulatory T cells and prevent inflammation [26]. The abundance of these obligate anaerobic bacteria in the intestine provides colonization resistance by reducing spatial opportunity for other organisms [27]. The organisms also produce antimicrobial substances such as bacteriocin [28], and competing for available nutrients [29]. Antibiotics that have activities against these obligate anaerobic bacteria, therefore, break down the colonization resistance and provide the opportunity for colonization by pathogenic bacterial species. Previous studies in patients who underwent allogeneic HSCT found that patients who exhibited loss of the genus *Blautia*, a Clostridial species, from the use of antibiotics that were active against obligate anaerobic bacteria were more likely to develop and die of severe GVHD [30]. More recently, the use of piperacillin/tazobactam, imipenem/cilastatin, and meropenem in patients who received chimeric antigen receptor T-cell (CAR-T) therapy for lymphoid malignancies was associated with higher risks for the development of immune cell-associated neurotoxic syndrome (iCANS) [31]. Both these highlight the importance of the intestinal obligate anaerobic bacteria in modulating inflammatory immune responses.

Based on the above consideration, our findings of increased risks for AGR after kidney transplant in patients who were exposed to PM within two months of transplant or during the early period after transplant are not surprising. It is likely that the reduction of the intestinal obligate anaerobic bacteria induced by piperacillin/tazobactam or meropenem, a carbapenem class of antibiotics, given prior to transplant or in the early period after transplant created a pro-inflammatory immune environment that renders the patients at risk for the development of AGR. Antibiotics with a narrower spectrum of activities should, therefore, always be used preferentially. Our data also suggest the need to have strict institutional guidelines for the use of antibiotics in patients before and after kidney transplant to minimize such an actionable factor for the development of AGR.

Our findings suffer the limitations associated with being a retrospective study. Since this is a medical record-based retrospective study, it is possible that not all the data, especially some potential confounding factors, was not accurately captured in the medical records. Furthermore, we have not factored into our analysis any possible effects of the differences in the duration of PM use, the effects of combination antibiotics, and the use of antibiotics beyond the pre-transplant or peri-transplant period in influencing the risks for the development of AGR. We also have not taken into consideration any possible differences in the trough levels of tacrolimus in the patients. Since only 18% of our patients received living donor kidneys, the number did not allow us to investigate any potential interaction between the use of PM and the method of graft harvesting, although a previous systemic review in living donor kidney transplant did not find any significant difference in the graft outcome between using laparoscopic donor nephrectomy and robot-assisted laparoscopic donor nephrectomy [32]. Not unlike a previous review paper that highlighted the impact of COVID-19 infection on the kidney transplant activity and the development of acute renal failure [33], the number of patients in our cohort was dramatically reduced in 2020, the last of the five years of activity we have included in our study. Data on any other viral infections also were not captured. As a result, any effect COVID-19 or other viral infections may have on the AGR on our patients could not be established.

In conclusion, exposure to PM within two months prior to and during the peri-transplant period after kidney transplant may be associated with higher risks for AGR. Our findings, however, need to be validated by an external cohort of kidney transplant patients and then confirmed prospectively. If confirmed, PM use in this group of patients should be kept to the minimum.

## Figures and Tables

**Figure 1 jcm-11-02726-f001:**
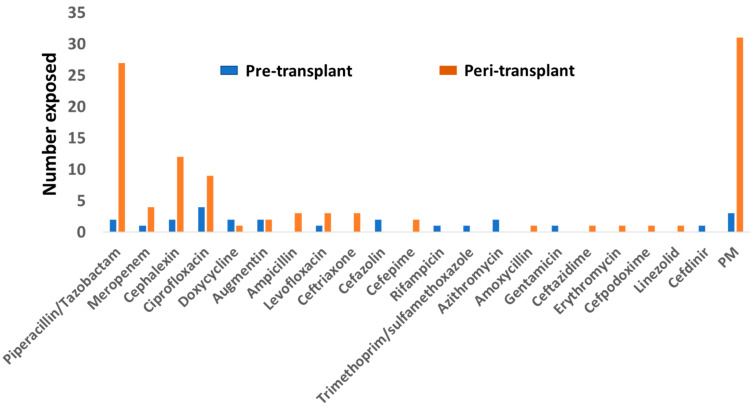
Antibiotics used on the patients within two months of transplant or during the peri-transplant period. The top three antibiotics were piperacillin/tazobactam, cephalexin, and ciprofloxacin. (PM = piperacillin/tazobactam + meropenem).

**Figure 2 jcm-11-02726-f002:**
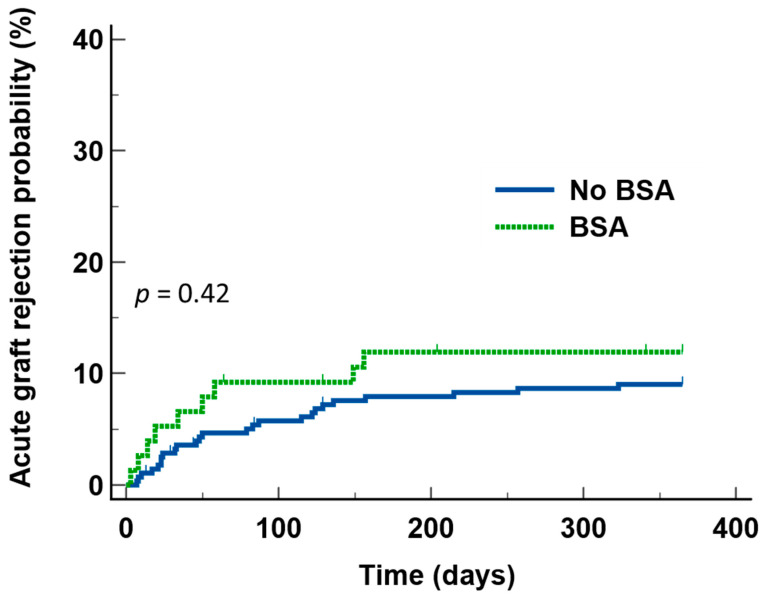
Acute graft rejection during the first year after transplant in patients who received broad-spectrum antibiotics (BSA) and those who did not, showing that there was not any statistical difference between the two groups.

**Figure 3 jcm-11-02726-f003:**
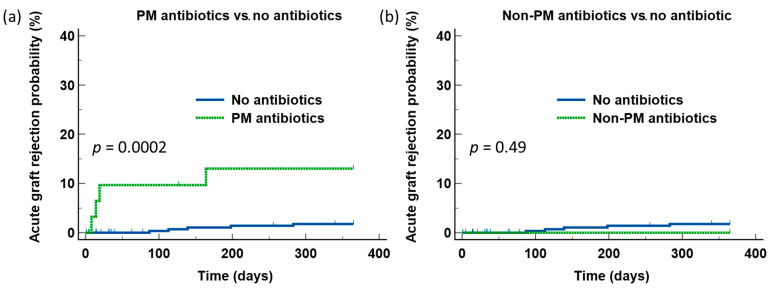
Comparison of the risks for acute graft rejection between patients who received PM and those who received non-PM antibiotics, with patients who did not receive any antibiotics in addition to the one dose of cefazolin in the operating room and PJP prophylaxis. Compared to those who did not receive any additional antibiotics, patients who received piperacillin/tazobactam or meropenem (PM) were at significantly higher risks for acute graft rejection during the first year after transplant (**a**). In contrast, such risks were not observed in patients who received non-PM antibiotics (**b**).

**Figure 4 jcm-11-02726-f004:**
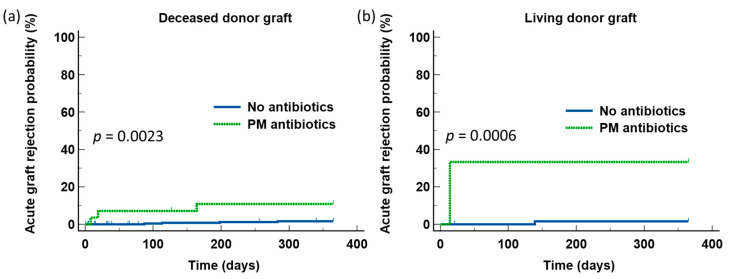
The detrimental effect of piperacillin/tazobactam or meropenem was observed in all the patients, irrespective of whether they received deceased donor kidney grafts (**a**) or living donor kidney grafts (**b**).

**Table 1 jcm-11-02726-t001:** Characteristics of patients.

	No Antibiotic (*n* = 286)	Antibiotics (*n* = 71)	*p* Value
**Age (years)**			
Median	53	50	*N.S.*
Range	7–76	13–74	
**Gender (F:M)**	117:169	26:45:00	*N.S.*
**Median Follow-up (days)**	1269.5	1270	*N.S.*
**Number of HLA mismatch**			
6/6 antigens	51 (17.5%)	15 (21.1%)	*N.S.*
5/6 antigens	100 (35.1%)	23 (32.4%)	*N.S.*
4/6 antigens	57 (20%)	13 (18.3%)	*N.S.*
3/6 antigens	40 14%)	11 (15.5%)	*N.S.*
2/6 antigens	16 (5.6%)	5 (7%)	*N.S.*
1/6 antigens	2 (0.7%)	0 (0%)	*N.S.*
0/6 antigens	20 (7%)	4 (5.6%)	*N.S.*
**Kidney donor source**			
Deceased donor	226 (79%)	64 (90%)	0.04
Living donor	60 (21%)	7 (10%)	

F:M = Female:Male; *N.S*. = not significant; HLA = Human Leukocyte Antigen.

**Table 2 jcm-11-02726-t002:** Indications for the use of broad-spectrum antibiotics.

Pre-Transplant (*n*)	Immediately Post-Transplant (*n*)
Urinary tract infection (7) Soft tissue infection (5) Bacteremia (1) Pneumonia (1) Latent tuberculosis (1) Bacteruria (1) Asthma exacerbation (1) Sinusitis (1) Upper respiratory infection (1)	Bacteruria (18) Donor bacteruria (8) Donor bacteremia (8) Soft tissue infection (7) Bacteremia/septicemia (4) Contaminated organ (4) Pneumonia (3) History of infective endocarditis (1) Peritonitis (1) Donor positive bronchial washing (1)

**Table 3 jcm-11-02726-t003:** Characteristics of patients with PM exposure vs. no PM exposure.

	PM Exposure (*n* = 32)	No PM Exposure (*n* = 325)	*p* Value
**Age (years)**			
Median	54	51	*N.S.*
Range	13–70	7–76	
**Gender (F:M)**	14:18	128:197	*N.S.*
**Number of HLA mismatch**			
6/6 antigens	8 (25%)	57 (17.5%)	*N.S.*
5/6 antigens	11 (34.4%)	112 (34.5%)	*N.S.*
4/6 antigens	6 (18.8%)	64 (19.7%)	*N.S.*
3/6 antigens	3 (9.4%)	48 (14.8%)	*N.S.*
2/6 antigens	2 (6.3%)	19 (5.8%)	*N.S.*
1/6 antigens	0 (0%)	2 (0.6%)	*N.S.*
0/6 antigens	2 (6.3%)	23 (7.1%)	*N.S.*
**Kidney donor source**			
Deceased donor	29 (90.6%)	261 (80.3%)	*N.S.*
Living donor	3 (9.4%)	64 (19.7%)	

F:M = Female:Male; *N.S*. = not significant; HLA = Human Leukocyte Antigen.

## Data Availability

Not applicable due to patient confidentiality.

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
