# Peer review of "Piperacillin/Tazobactam and Meropenem Use Increases the Risks for Acute Graft Rejection Following First Kidney Transplantation"

_jcm, 2022, doi:10.3390/jcm11102726_

Round 1

Reviewer 1 Report

Authors of the present study aimed at testing the hypothesis that BSA use shortly before and soon after transplant may affect the risks for the development of acute graft rejection (AGR) following kidney transplant.

A retrospective cohort study was performed on all patients who underwent kidney transplants from January 1, 2016 to December 31, 2020. Overall, 357 consecutive patients (143 females and 214 males) were included in the study. There was no difference in the AGR rates during the first year between patients who received BSA and those who did not. However, the use of piperacillin/tazobactam or meropenem (PM) was associated with increased risks for the development of AGR, irrespective of the source of the donor grafts.

The aim of the study is original. However, I suggest some revisions to improve the overall quality of the study:

  1. Methods: describe how continuous and categorical variables are reported, describe statistical software used.

2: Results: Any information about the duration of antibiotic therapy? Combination therapies? Which criteria for the selection of antibiotics in your center? Any information about graft harvesting and its potential impact on outcomes? (Creta M, Calogero A, Sagnelli C, Peluso G, Incollingo P, Candida M, Minieri G, Longo N, Fusco F, Tammaro V, Dodaro CA, Mangiapia F, Carlomagno N. Donor and Recipient Outcomes following Robotic-Assisted Laparoscopic Living Donor Nephrectomy: A Systematic Review. Biomed Res Int. 2019 Apr 22;2019:1729138. doi: 10.1155/2019/1729138. PMID: 31143770; PMCID: PMC6501265.)

  1. Discussion: Are there specific guidelines for antibiotic therapy in kidney transplant patients?
  2. The potential role of viral infections should be discussed (See this interesting paper: Sagnelli C, Sica A, Gallo M, Peluso G, Varlese F, D'Alessandro V, Ciccozzi M, Crocetto F, Garofalo C, Fiorelli A, Iannuzzo G, Reginelli A, Schonauer F, Santangelo M, Sagnelli E, Creta M, Calogero A. Renal involvement in COVID-19: focus on kidney transplant sector. Infection. 2021 Dec;49(6):1265-1275. doi: 10.1007/s15010-021-01706-6. Epub 2021 Oct 5. PMID: 34611792; PMCID: PMC8491762).
  3. A more accurate discussion about all potential limits of the study as well as future directions would be of interest.
  4. A “Conclusion” section would be of interest.

Author Response

REVIEWER #1

Methods: describe how continuous and categorical variables are reported, describe statistical software used.

RESPONSE: We have now clarified both in the revised manuscript.

Results: Any information about the duration of antibiotic therapy?

RESPONSE: We did not have the exact duration of antibiotic therapy. However, all the patients received at least three days of the antibiotics. We have added this point to the revised manuscript under the Limitations of the study.

Combination therapies? Which criteria for the selection of antibiotics in your center?

RESPONSE: We accept these two points and have addressed them under the Limitations of the study.

Any information about graft harvesting and its potential impact on outcomes? (Creta M, Calogero A, Sagnelli C, Peluso G, Incollingo P, Candida M, Minieri G, Longo N, Fusco F, Tammaro V, Dodaro CA, Mangiapia F, Carlomagno N. Donor and Recipient Outcomes following Robotic-Assisted Laparoscopic Living Donor Nephrectomy: A Systematic Review. Biomed Res Int. 2019 Apr 22;2019:1729138. doi: 10.1155/2019/1729138. PMID: 31143770; PMCID: PMC6501265.)

RESPONSE: Since the majority of the patients received deceased donor kidneys, we did not have information on the graft harvesting methods. Because of that, we have addressed this shortcoming in the Limitations of the study.

Discussion: Are there specific guidelines for antibiotic therapy in kidney transplant patients?

RESPONSE: There is not any specific guidelines for antibiotic therapy in kidney transplant patients. Our study highlights the need for such guidelines. We have addressed such a need in the Discussion of the revised manuscript.

The potential role of viral infections should be discussed (See this interesting paper: Sagnelli C, Sica A, Gallo M, Peluso G, Varlese F, D'Alessandro V, Ciccozzi M, Crocetto F, Garofalo C, Fiorelli A, Iannuzzo G, Reginelli A, Schonauer F, Santangelo M, Sagnelli E, Creta M, Calogero A. Renal involvement in COVID-19: focus on kidney transplant sector. Infection. 2021 Dec;49(6):1265-1275. doi: 10.1007/s15010-021-01706-6. Epub 2021 Oct 5. PMID: 34611792; PMCID: PMC8491762).

RESPONSE; We agreed with the reviewer and have included the potential role of viral infections in our discussion.

A more accurate discussion about all potential limits of the study as well as future directions would be of interest.

RESPONSE: We accept the reviewer’s comments and have added discussions of the above issues in the revised manuscript.

A “Conclusion” section would be of interest.

RESPONSE; We have not added a Conclusion section since this is not the format of the journal. However, we have included a paragraph in our revised Discussion to serve as the Conclusion.

Reviewer 2 Report

Surprising results.  Please have the best writer in your group go over the paper line by line (including the abstract) and make a small number of minor changes to the grammar to bring this paper up to standard.  For example, the first sentence of the abstract needs "the" prior to "intestinal". 

Author Response

REVIEWER #2

Surprising results.  Please have the best writer in your group go over the paper line by line (including the abstract) and make a small number of minor changes to the grammar to bring this paper up to standard.  For example, the first sentence of the abstract needs "the" prior to "intestinal".

RESPONSE: The revised manuscript has undergone extensive grammatical check.

Round 2

Reviewer 1 Report

The authors answered all comments and suggestions.